# Current Issues within the Perinatal Mental Health Care System in Aichi Prefecture, Japan: A Cross-Sectional Questionnaire Survey

**DOI:** 10.3390/ijerph18179122

**Published:** 2021-08-29

**Authors:** Kei Fujita, Tomomi Kotani, Yoshinori Moriyama, Takafumi Ushida, Kenji Imai, Tomoko Kobayashi-Nakano, Noriko Kato, Takeo Kano, Fumitaka Kikkawa, Hiroaki Kajiyama

**Affiliations:** 1Department of Obstetrics and Gynecology, Nagoya University Graduate School of Medicine, 65 Tsurumai-Cho, Showa-Ku, Nagoya 466-8550, Aichi, Japan; kfujita620@yahoo.co.jp (K.F.); yoshinori.moriyama@fujita-hu.ac.jp (Y.M.); u-taka23@med.nagoya-u.ac.jp (T.U.); kenchan2@med.nagoya-u.ac.jp (K.I.); tnakano@med.nagoya-u.ac.jp (T.K.-N.); kikkawaf@med.nagoya-u.ac.jp (F.K.); kajiyama@med.nagoya-u.ac.jp (H.K.); 2Department of Obstetrics and Gynecology, Anjo Kosei Hospital, 28 Higashihirokute, Anjo-Cho, Anjo 446-8602, Aichi, Japan; 3Division of Perinatology, Center for Maternal-Neonatal Care, Nagoya University Hospital, Nagoya 466-8560, Aichi, Japan; 4Department of Obstetrics and Gynecology, Fujita Health University School of Medicine, Toyoake 470-1192, Aichi, Japan; 5Department of Obstetrics and Gynecology, Japanese Red Cross Aichi Medical Center Nagoya Daini Hospital, 2-9 Myoken-Cho, Showa-Ku, Nagoya 466-8650, Aichi, Japan; noripi@nagoya2.jrc.or.jp; 6Kano’s Clinic for Women, 3-16-25 Osu, Naka-Ku, Nagoya 460-0011, Aichi, Japan; kanotake@mub.biglobe.ne.jp

**Keywords:** psychotropic drugs, multidisciplinary, preconception

## Abstract

Mental illnesses commonly occur in the reproductive age. This study aimed to identify the issues that exist within the perinatal mental health care system. A cross-sectional survey was conducted in Aichi Prefecture in central Japan. Questionnaires on the situation between 2016 and 2018 were mailed to the head physicians of 128 maternity care units, 21 neonatal intensive care units (NICUs), and 40 assisted reproductive technology (ART) units. A total of 82 (52.6 per 100,000 births) women were admitted to mental health care units during the perinatal period, and 158 (1.0 per 1000 births) neonates born to mothers with mental illness were admitted to NICUs. Approximately 40% of patients were hospitalized in psychiatric hospitals without maternity care units. Eighty-four (71.1%) and 76 (64.4%) maternity care units did not have psychiatrists or social workers, respectively. Moreover, 20–35% of the head physicians in private clinics, general hospitals, and ART units endorsed the discontinuation of psychotropic drug use during pregnancy. However, the corresponding figures were only 5% among those in maternal-fetal centers. Resources for perinatal mental illness might be limited. Perspectives on psychotropic drug use differed based on the type of facilities where the doctors were working.

## 1. Introduction

Approximately 10–20% of women worldwide experience mental illness during the perinatal period [1]; growing evidence suggests that mental illness has adverse effects on mothers and their children [2,3,4]. During the perinatal period, women with mental illness should receive care from a multidisciplinary team that includes psychiatrists, obstetricians, midwives, psychologists, and social workers [5]. Further, women with mental illness should be in collaboration with psychiatric service providers during the perinatal period [6]. However, more research is expected to establish robust perinatal mental health care [7]. 

In Japan, the maternal suicide rate in Tokyo’s 23 wards was remarkably high, at 8.7/100,000 births, from 2005 to 2014 [8], when compared with the suicide rates of 1.2 and 2.0/100,000 births in Italy [9] and the USA [10], respectively. In response to this, the Japanese government announced a series of new administrative policies. In April 2016, the Japanese Ministry of Health, Labor, and Welfare added pregnant and postpartum women with mental illness to the list of eligible patients for calculating an additional fee for managing in- and outpatients. In April 2017, it established the “Guidelines for the operations of child-rearing generation comprehensive support centers”, suggesting that pregnant and postpartum women with mental illness should be supported by a multidisciplinary team. The team discusses the medical and social problems the women face with the patients and their family members and provides childcare and financial or mental support, as necessary. The team also seamlessly collaborates with hospitals, clinics, and public health centers of their community. Public health nurses in public health centers provide outreach services by visiting patients’ homes, as required. Simultaneously, financial support for screening using the Edinburgh Postnatal Depression Scale at postnatal visits has also been promoted by the Japanese government and initiated in some municipalities. In April 2018, management as a multidisciplinary team was partially financially supported. 

The Japanese organization of perinatal services and current health policies affecting perinatal women and their families are as follows: private obstetric clinics provide most maternal care. Assisted reproductive technology (ART) is also provided by private clinics. Women with obstetric or non-obstetric complications are both emergently and non-emergently referred to maternal-fetal (MF) centers in the perinatal community system. MF centers have NICU. However, women with mental illnesses were not considered until the publication of the report from Tokyo’s 23 wards. Thus, some MF centers have mental health care units that can hospitalize patients emergently. In Japanese customs, women often return to their hometown and deliver their babies in hospitals or clinics near their parents’ homes. Their mothers often help their daughters take care of their babies. However, with an increase in older pregnancies, grandmothers become more elderly and find it challenging to help their daughters. Women can consult at a public “child-rearing generation comprehensive support center” in their communities and receive “postpartum care services” as an accommodation-type or day-care type, as required. In Aichi, accommodation-type postpartum care is available in several private clinics. Women without support from their families can stay in these clinics for several weeks after leaving the hospital where they delivered. Social workers help women receive these services by collaborating with community health care providers. 

However, whether the current system works effectively remains unknown. Therefore, it is necessary to analyze the current situation in maternal mental health care, including examining medical resources such as psychiatrists and social workers. This is needed to deepen the knowledge of several organizational protocols present in Aichi Prefecture and highlight the urgent need for changes in local and national health policies. In addition, most mental health care units are placed in specialized psychiatric hospitals that do not include wards for peri- or postnatal women. Furthermore, the number of hospitals with maternity care units and mental health care units is limited in Aichi Prefecture. Thus, it is necessary to clarify where pregnant or postpartum women with deteriorating mental illnesses are hospitalized. 

Psychotropic drug use is a problem in perinatal mental health care. One report states that 7.1% of women use psychotropic medication during pregnancy in Canada [9]. The risk of major depression relapse is substantially higher after the discontinuation of drug intake during pregnancy [10]. A recent systematic review also concluded that discontinuing psychotropic drug treatment increases the risk of relapse, recurrence, and suicide [11]. However, there is insufficient literature on the assessment of the risk–benefit ratio for psychotropic drug use during pregnancy [12,13] because randomized controlled trials have not been conducted owing to ethical issues. Clinicians are required to make decisions on drug use based on the results of animal studies and retrospective studies on short-term pregnancy outcomes, including teratogenicity and low birth weight [11]. However, literature on the long-term neurobehavioral consequences of drug use is limited, and the safest psychotropic drugs for pregnant women are yet to be identified [13]. Therefore, decisions about psychotropic drug use and its discontinuation during pregnancy should be carefully made on a case-by-case basis [13]. The perspectives of obstetricians who provide maternity care would significantly affect the decision-making process for their patients. If more obstetricians considered discontinuing psychotropic drugs during pregnancy, the risk of relapse by discontinuation in Aichi Prefecture should be discussed. Thus, it is meaningful to clarify the perspectives of obstetricians on psychotropic drug use or its discontinuation during pregnancy. Furthermore, it would be helpful to determine the neonatologists’ policies in improving the management of infants born to mothers with mental illness because neonatal abstinence syndrome is known in intrauterine exposure to psychotropic drugs [9].

It has also been recommended that one minimize the number of drugs that are consumed and switch to safer drugs (if possible) “before conception” [14]. Appropriate planning and intervention before pregnancy (e.g., preconception counseling, including medication) can improve the outcomes [14]. Mental health should also be considered as an important preconception health indicator [15]. Preconception and public health strategies are believed to have the greatest impact on the population’s health [1]. The new perinatal community-based mental health services in England provide preconception counseling to all referred women with moderate to severe mental illnesses who are planning a pregnancy [16]. Fertility therapy allows one to support planned pregnancy and provide preconception counseling to women with mental illnesses in collaboration with psychiatrists. However, how preconception care is implemented for women with mental illness during fertility treatment in Japan remains unknown. Therefore, investigating their collaboration with psychiatrists before conception and the perspectives on drug use is important among doctors who work in ART units. 

This study surveys the perinatal mental health care system in Aichi Prefecture, Japan, including women with worsening mental illnesses during the perinatal period, multidisciplinary perinatal mental health care system, perspectives of doctors on psychotropic drug use during the perinatal period, NICU admission of infants born to mothers with mental illness, and preconception counseling provided by ART units. 

## 2. Materials and Methods

### 2.1. Setting

We conducted a cross-sectional survey across medical facilities in Aichi Prefecture in central Japan. A total of 187,892 babies were born in Aichi Prefecture between 2016 and 2018, accounting for 6.6% of live births in Japan during this period (*n* = 2,841,788; The Japan Ministry of Health, Labor, and Welfare Vital Statistics of Japan; https://www.mhlw.go.jp/toukei/list/81-1a.html (accessed on 26 August 2021). Aichi’s perinatal outcomes ranking, among all prefectures (*n* = 47), in 2016, 2017, and 2018 was as follows: birth rates, 2nd, 3rd, and 2nd; perinatal mortality rates, 22nd, 39th, and 35th, respectively. In 2019, there were 128 maternity care units in the prefecture, including 20 MF centers, 19 general hospitals, and 89 private clinics (Figure 1). 

### 2.2. Design

Questionnaires were mailed to the head physicians, as directors or heads of units, of all the 128 maternity care units (Table A1), 21 NICUs (Table A2), and 40 ART units (Table A3) in Aichi, between April and December 2019. Contents in the questionnaire were also discussed in the Perinatal Care Association of the Aichi Prefectural Government before drafting. 

The questionnaire comprised nine close-ended questions and one open-ended question for maternity care units. In the first part (Table A1, Q1-1 to 1-4), the multidisciplinary care system was investigated, such as the number of psychiatrists or social workers working in the same facility; the collaboration of psychiatrists or social workers working in the other facilities, if unavailable; policies of management of women with mental illness; and the experience of collaboration with the public health center. In the second part (Table A1, Q2-1 to 2-4), the number of hospitalization cases was investigated from 2016 to 2018. In the third part (Table A1, Q3), opinions on psychotropic drug use during pregnancy were elicited. In the last part, opinions on the system in Aichi Prefecture were collected (Table A1, Q4). The policies of infant care (Table A2) and preconception care during fertility treatment (Table A3) were also investigated. The following details were also collected: the number of deliveries and patients admitted to mental health care units during the perinatal period (Table A1); the number of neonates born to mothers with mental illnesses and admitted to NICUs (Table A2), from 2016 to 2018; and the number of in vitro fertilization (IVF), intracytoplasmic sperm injection (ICSI), and frozen embryo transfer (FET) treatment cycles in 2017 (Table A3. Mental health conditions included the following: common mental disorders, severe mental disorders, and substance use disorders. 

### 2.3. Statistical Analysis

Statistical analysis was conducted using JMP Pro 15 (JMP Japan, Tokyo, Japan). We summarized responses to each question by computing frequencies and percentages. The different types of medical care units, namely MF centers, general hospitals, and private clinics, were compared. Temporal (yearly) changes in the rate of admission to mental health care units were also examined. Categorical variables were examined using Fisher’s exact test. Statistical significance was assessed at *p*  <  0.05.

## 3. Results

### 3.1. Response Rates

A total of 118 maternity care units (92.2%) responded and provided the number of deliveries. Only ten private clinics failed to respond. All the MF centers and general hospitals responded to the questionnaire. A total of 155,417 infants were born in these maternity care units between 2016 and 2018, and 102,027 (66.7%) babies were born in private clinics. Further, 19 of the 21 NICUs (90.5%) and 28 of the 40 ART units (70.0%) responded. The NICUs that participated in this survey had a total of 173 beds, with the median number of beds in each NICU being 9 (range = 3–18) (Table A2, Q1).

### 3.2. Women with Worsening Mental Illness during the Perinatal Period

A total of 82 women (52.6 per 100 000 births) were admitted to mental health care units during the perinatal period (Table A1, Q2-1). There was a significant increase in the percentage of admissions across time: the number of admissions (*n* = 38) in 2018 (73.6 per 100 000 births) was approximately twice that in 2016 (*n* = 19; 36.0 per 100,000 births) (Figure 2, *p* = 0.01). 

Moreover, of 82 patients, 32 (39.0%) were referred to specialized psychiatric hospitals without maternity care units for hospitalization. The others were admitted to the mental health care units of general hospitals with maternity care units (Table A1, Q2-2). Despite a deterioration in their mental illness, six patients were emergently admitted to maternity care units because they could not be admitted to mental health care units (Table A1, Q2-3). Additionally, ten patients had undergone legal abortions because of worsening mental illness across the three years of the survey (Table A1, Q2-4).

### 3.3. Multidisciplinary Perinatal Mental Health Care System

#### 3.3.1. Psychiatrists

Most MF centers and general hospitals had psychiatrists. The distributions were significantly different across units: MF centers, general hospitals, and private clinics (Table A1, Q1-1, Figure 3a, *p* < 0.01). 

Most maternity care units (*n* = 84, 71.0%) did not have psychiatrists (Figure 4a). Only six units (5%) could manage emergency admissions (Figure 4b), of which only four were MF centers. Further, only 12 units (14.3%) admitted having contacted their patients’ psychiatrists from other medical facilities and providing care during the perinatal period (Table A1, Q1-1). Fifty-two units (61.2%) replied that they referred those patients to MF centers or general hospitals (Table A1, Q1-2). Additionally, 28 units without psychiatrists (33.3%) sometimes provided maternity care to patients who were stable or/and did not require medication (Table A1, Q1-2). 

#### 3.3.2. Social Workers

Most MF centers and general hospitals had hospital social workers, who were also distributed significantly different across the different types of units (Table A1, Q1-3, Figure 3b, *p* < 0.01). The 76 maternity care units were without any social workers. Further, only 12 units (16.0%) indicated their collaboration with community social workers as an alternative. Concerning seamless multidisciplinary collaboration between medical units and public health centers, 58 maternity care units (49.0%) responded that they encountered cases requiring collaboration with a public health center. The corresponding percentages did not differ significantly across units (Table A1, Q1-4, Figure 3c, *p* = 0.549). The availability of social workers in these units was unrelated to the experience of seamless multidisciplinary collaboration (unavailable vs. available; 52.2% vs. 60.6%, *p* = 0.525). Among the units that collaborated with the public health center, 57 (98.3%) answered that this approach was adopted seamlessly (Table A1, Q1-4).

### 3.4. Problems within the Perinatal Mental Health Care System

Respondents (21/118) provided a range of opinions in response to the following question (Table A1, Q4): “What problems exist within the present perinatal mental health care system in Aichi Prefecture?” Eleven respondents reported that the available perinatal mental health care resources were inadequate to permit them to refer patients emergently when their mental illness worsened. The other shared opinions included: patients rejecting referrals to psychiatrists; poor medication adherence; and a lack of consensus regarding therapeutic protocols between obstetricians and psychiatrists.

### 3.5. Perspectives on Psychotropic Drug Use during Pregnancy among Obstetricians

Table 1 summarizes the participating obstetricians’ perspectives on psychotropic drug use when non-contraindicated psychotropic drugs during pregnancy are prescribed (Table A1, Q3). This variable was significantly associated with the unit type (*p* = 0.033). In addition, 5.0%, 31.6%, and 35.4% of MF centers, general hospitals, and private clinics, respectively, endorsed the discontinuation of drug use. 

### 3.6. NICU Admission of Infants Born to Mothers with Mental Illness

Among the 19 participating NICUs, a total of 158 infants (1.0 per 1000 births; 50, 53, and 55 infants in 2016, 2017, and 2018, respectively) born to mothers with mental illness were admitted to NICUs during the survey period (Table A2, Q5). The policies of NICUs are shown in Table A2 (Q2-4). 

### 3.7. Preconception Counseling Provided by ART Units

A questionnaire was sent to ART units to survey the preconception counseling services provided by fertility therapy centers (Table A3). The responded ART units (*n* = 28) recorded 3921 IVF, 7624 ICSI, and 13,192 FET treatment cycles in 2017 (Table A3, Q1), which accounted for approximately 4.3%, 4.8%, and 6.6% of the total treatment cycles in Japan, respectively. 

The policies and perspectives of ART units are shown in Table A3, Q2-5. 

Perspectives regarding psychotropic drug use during pregnancy were also examined (Table A3, Q7), with six ART units (21.4%) endorsing drug use discontinuation. 

Ten respondents provided a range of opinions in response to the following question (Table A3, Q8): “What problems have you encountered until now, when providing fertility treatment to women with a history of mental illness?”, including insufficient perinatal mental health care resources (*n* = 2); patient rejection of referrals to psychiatrists (*n* = 2); a lack of consensus regarding therapeutic protocols between fertility doctors and psychiatrists (*n* = 2); and insufficient information about prior history of mental illness because it was provided by clients (*n* = 2).

## 4. Discussion

### 4.1. Main Findings

This survey is the first to examine obstetricians’ perspectives on psychotropic drug use among pregnant women in Aichi Prefecture, Japan. Among the surveyed obstetricians in private clinics, 35.4% endorsed drug use discontinuation. However, 90.0% of those in MF centers endorsed the continuation of drug use. Their perspectives differed based on the type of maternity care unit in which they were working. Concerning fertility doctors, 21.4% of ART units endorsed drug use discontinuation.

In this study, several problems within the perinatal mental health care system in Aichi Prefecture, Japan, were identified. First, the number of admissions to mental health care units was 52.6 per 100,000 births, a figure that is increasing over time. Some patients were admitted to inappropriate units, such as psychiatric hospitals without maternity care units or maternal care units that are not equipped to provide specialized treatment for mental illnesses. In response to an open-ended question, some maternity care and ART units pointed out a lack of perinatal mental health care resources to which patients can be referred. Second, 71.0% and 64.4% of the maternity care units were without psychiatrists and social workers, respectively. The availability of these resources was significantly associated with the type of maternity care units, and most private clinics did not have psychiatrists or social workers. Third, the number of neonates in NICUs born to mothers with mental illness was 1.0 per 1000 births. However, a few of them were evaluated with neonatal toxicity caused by prenatal exposure. Finally, almost 40.0% of the ART units answered that they had no information about hospitals that provide specialized perinatal mental health care. Further, 67.9% of the ART units reported that no discussion with patients and their partners took place where their patients could receive appropriate care during the perinatal period before providing fertility treatment.

### 4.2. Interpretation of Main Findings

A total of 102,027 babies (66.4%) were born in private clinics. The high rate of childbirths in private clinics is a distinguishing characteristic of the perinatal care system in Japan. These results underscore a potential risk factor. Specifically, many women with mental illnesses may be discouraged from continuing psychotropic drug use when they are pregnant. Physicians in private clinics and ART units appeared to be more attuned to the potential risks of prenatal psychotropic medication exposure than those in MF centers, which endorsed the continued use of these drugs during the perinatal period (if necessary). These perspectives might depend on whether these units have psychiatrists and social workers. Most private clinics are not equipped with these resources. This decision-making process of drug use necessitates effective communication between psychiatrists, fertility doctors, obstetricians, and discussions with patients and patients’ family members in a multidisciplinary team. In this survey, only 14.3% of maternity care units without psychiatrists contacted their patients’ psychiatrists from other medical facilities. The government has promoted a multidisciplinary team, but it might not yet have reached a point of collaboration among different facilities, although it can be considered the beginning of a change. Additionally, this collaborative discussion should be initiated when a patient expresses her wish to conceive. However, the present findings indicate that the provision of such interventions in ART units is limited. Therefore, such patients might consult with perinatal mental health care experts before conception.

The findings of this study suggest that the available professional perinatal mental health care resources in Aichi Prefecture, Japan, are insufficient. Patients are often admitted to inappropriate units. In this survey, approximately 40% of patients were transferred to psychiatric hospitals without maternity care units, suggesting that mothers are separated from their babies and not cared for by obstetric staff, including midwives. Additionally, no particular attention was paid to neonates born to mothers with mental illnesses, even though the adverse effects of maternal mental illness on neonates are well documented. In several countries in Europe and South Asia, the USA, and Australia, “psychiatric mother-baby units” have been viewed as best practices for critical patients [1,17,18]. However, these have not yet been introduced in Japan. The present study also shows a significant increase in the percentage of admissions. From these findings, establishing psychiatric mother-baby units should be considered in Japan, although the cost-effectiveness and benefits in the country remain unclear [19]. However, even in those countries with psychiatric mother-baby units, ensuring that specialist perinatal mental health services are sufficiently resourced to treat women with mental health problems quickly is thought to be a challenging enough target [1]. Thus, as an alternative, more intervention in preconception [16], providing care involving patients’ family members [20], or increasing specialists for perinatal mental health care [1], should be considered, which could decrease inpatient-care needs by preventing deterioration during the perinatal period. In a single-center study, a comprehensive treatment as a multidisciplinary team has been reported to decrease the rate of patients’ deterioration or relapse of mental disorders in Japan [21]. Thus, improving the quality of a multidisciplinary team is also necessary.

### 4.3. Strengths and Limitations

The present findings delineate obstetricians, neonatologists, and fertility doctors’ perspectives on managing maternal mental illnesses in Aichi Prefecture, Japan. First, the response rates were as high as 90%, and the survey was conducted across all types of maternity care units. These factors possibly minimized non-response bias for the data based on the perspectives. Second, the treatment policy of neonatologists was also collected in mothers with mental illness. Third, information about preconception care and counseling was collected from fertility doctors. This study is the first survey to have examined fertility doctors’ perspectives on preconception care and counseling for women with mental illnesses in Aichi Prefecture, Japan.

The limitations of this study include the adoption of a questionnaire survey design. First, the clinical data of patients with mental illnesses were not collected. Concerning neonates born to mothers with mental illnesses, their prognosis could not be determined; further research is needed to bridge this gap in the literature. We did not determine whether NICU admission was directly necessitated by maternal mental illness. However, studies have found that NICU admission rates are higher among mothers with mental illnesses [17,18]. Second, it is difficult to generalize the results of this study for the whole of Japan, including in terms of the availability of psychiatrists and other resources. The birth rate in Aichi Prefecture is the second or third highest in Japan, and the live births in Aichi accounted for 6.6% of Japan’s. In contrast, the prefecture was ranked in the middle in the perinatal mortality rate, suggesting that the perinatal care system in Aichi Prefecture might be considered average in the country. However, Aichi Prefecture is located in the center of Japan (Figure 1) and is convenient for transportation; remote areas may have fewer resources. A nationwide survey is required in the future to solve this problem. Third, the service by a multidisciplinary team was evaluated only through a survey based on obstetricians’ perspectives. A survey among public health nurses and women with service experience should also be conducted to evaluate whether the system works effectively. We intend to survey public health nurses, although this has recently been reported in Ireland [22]. Fourth, the system currently seems to change along with these policies in Japan. From April 2016 to April 2018, the Japanese government launched a series of new policies, including guidelines and financial resources. Thus, in the period of this survey, these policies were not reflected. In Switzerland, the clinical trial to identify the problem and reform the perinatal mental health care system has started recently [23]. It is crucial to identify issues related to the perinatal mental health care system and improve it in high-income countries. In the future, the Japanese system should also be re-evaluated. Evaluating the system would result in identifying the problems, which could consequently improve the system. 

## 5. Conclusions

More physicians in private clinics endorsed drug use discontinuation than those in MF centers, even if those drugs were not contraindicated during pregnancy. The available professional perinatal mental health care resources were perceived to be insufficient, and the preconception care provided by fertility treatment centers was limited. Some new policies, including psychiatric mother-baby units, should be established to provide professional care to women with mental illnesses who wished to conceive in Japan, although further research should be conducted in this regard.

## Figures and Tables

**Figure 1 ijerph-18-09122-f001:**
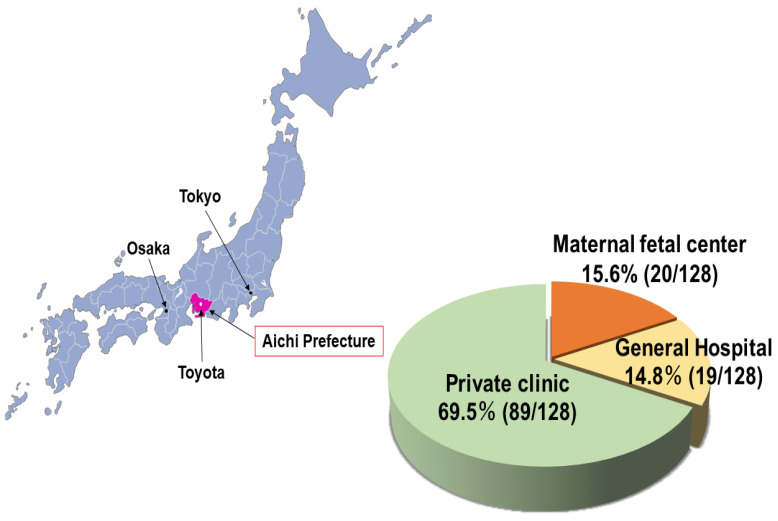
The system of maternity care units in Aichi Prefecture. Most maternity care was provided by private clinics. The map was created by Japanese Map Maker (https://www.start-point.net/maps/map_maker/Public Day (accessed on 8 October 2020 18:39:43) (last updated on 8 October 2020 20:29:51).

**Figure 2 ijerph-18-09122-f002:**
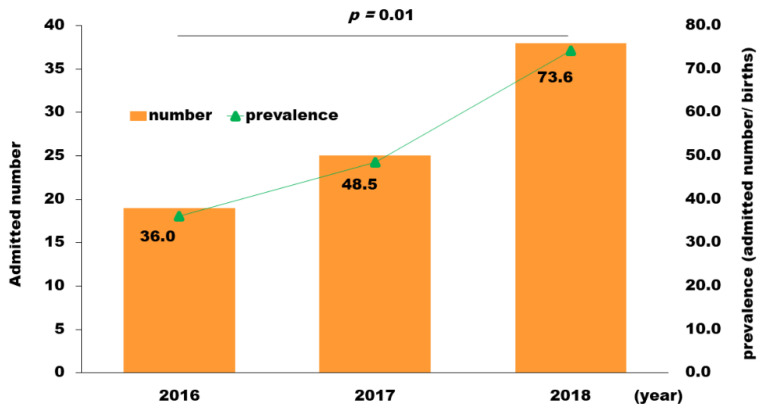
The annual number and prevalence of women admitted to mental health care units during the perinatal period. The number of admissions in 2018 (73.6 per 100,000 births) was approximately twice that in 2016 (36.0 per 100,000 births) (*p* = 0.01).

**Figure 3 ijerph-18-09122-f003:**
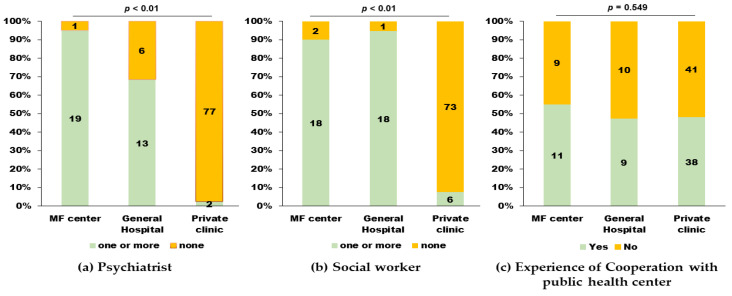
Multidisciplinary perinatal mental health care resources across different units: Maternal-fetal (MF) centers, general hospitals, and private clinics. (**a**) The distribution of psychiatrists. (**b**) The distribution of hospital social workers. (**c**) Experience of seamless multidisciplinary collaboration between medical units and community health centers.

**Figure 4 ijerph-18-09122-f004:**
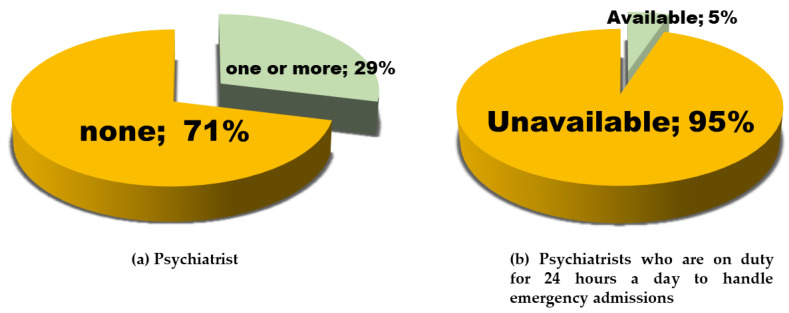
The availability of psychiatrists in maternal care units: (**a**)The percentage of maternity care units that have one or more psychiatrists or none. (**b**) The percentage of maternity care units with psychiatrists on duty 24 h a day to manage emergency admissions is shown.

**Table 1 ijerph-18-09122-t001:** Perspectives on non-contraindicated psychotropic drug use during pregnancy across maternal care units.

Type of Maternity Care Units	Psychotropic Drug Use during Pregnancy			*p*-Value
	Discontinue	Continue	Other	
MF center (*n* = 20)	1 (5.0)	18 (90.0)	1 (5.0)	0.033
General hospital (*n* = 19)	6 (31.6)	13 (68.4)	0 (0.0)	
Private clinic (*n* = 79)	28 (35.4)	45 (57.0)	6 (7.6)	
Total (*n* = 118)	35 (29.7)	76 (64.4)	7 (5.9)	

Values represent frequencies (percentages), and the *p*-value was computed using Fisher’s exact test. MF = maternal-fetal.

## Data Availability

The datasets used and analyzed during the current study are available from the corresponding author on reasonable request.

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
