# Peer review of "Current Issues within the Perinatal Mental Health Care System in Aichi Prefecture, Japan: A Cross-Sectional Questionnaire Survey"

_ijerph, 2021, doi:10.3390/ijerph18179122_

Round 1

Reviewer 1 Report

This manuscript deals with an interesting survey of doctors in Aichi Prefecture, Japan, that included respondents from Neonatal Intensive Care Units (NICU) and Assisted Reproductive Technology (ART) Units, and that aims to identify issues within the perinatal mental health system including perspectives on psychotropic drug use during the perinatal period.

In this first draft of the manuscript some critical issues emerge in the MATERIALS AND METHODS section. In order to be better appreciated also by readers outside Japan, the work needs further short additions within the INTRODUCTION section.

These are my suggestions regarding each single section.

INTRODUCTION

In the introductory section, more space should be given to the description of:

  • Japanese organization of perinatal services and current health policies affecting perinatal women and their families;
  • the role of Japanese public and private services in the perinatal area;
  • a description of local and hospital services on perinatality;
  • the configuration of NICU and ART teams in Japan;
  • the (possible) socio-cultural peculiarities and available services in Aichi Prefecture.

What led the authors to specifically investigate “…… psychotropic drug use during the perinatal period among doctors in Aichi Prefecture, Japan”.

Additionally, the description of the objectives is partial and does not adequately prepare the reader to face and understand the meaning of the results obtained.

MATERIALS AND METHODS

What is the professional profile of the 118 subjects who answered the questionnaire?

It should be made clear that the survey also aims to deepen knowledge of some organizational methods present in Aichi Prefecture and to urge the need for changes in local and national health policies.

INSTRUMENT

What criteria were adopted for the drafting of the questionnaire for maternity care units?

RESULTS

The topics statistically treated in each individual sub-section of “3.3. Multidisciplinary perinatal mental health care system” must be previously presented in the final part of the INTRODUCTION section to prepare the reader for the results obtained.

REFERENCES

Progressive numbers that refer to bibliographic citations are missing.

Reviewer 2 Report

Thanks you for this paper. It is a survey of a particular area of Japan and as such, is not a scientifically rigorous study. The findings are unlikely to be of interest to an international audience but will to be of interest in Japan. I think that it would be best directed to a Japanese journal.

In terms of content it is far too long for the quality of the data which is descriptive in nature. Much of the text could be summarised in one or 2 tables/charts with the outstanding results provided. The most obvious is that most of the maternity services do not have a psychiatrist involved. This very important figure is lost in the many aspects you present from each clinic type.

I note that you comment that the impact of the new approach to encouraging multidisciplinary assessment has shown no improvement but from a very low start, the numbers admitted have increased, so it may be the beginning of a change.  

Reviewer 3 Report

The paper provides interesting information on the issue of perinatal mental health care service in Aichi Prefecture, Japan. It is a well-written and useful summary of the current status of the art by the point of view of the care providers.

I have a single minor suggestion for the authors, who are to be congratulated for their work. The high rate of maternal suicide rate (8.7/100000 births) reported in Tokyo's 23 wards from 2005 to 2014 makes the issue of improvement of perinatal mental health care compelling in the Country. In order to make this point clearer to the international readership of the journal, I would suggest adding some data and references on maternal suicide rate in other countries: e.g.,  

  • Khalifeh H, Hunt IM, Appleby L, Howard LM. Suicide in perinatal and non-perinatal women in contact with psychiatric services: 15 year findings from a UK national inquiry. Lancet Psychiatry. 2016 Mar;3(3):233-42. doi: 10.1016/S2215-0366(16)00003-1. Epub 2016 Jan 16. PMID: 26781366.
  • Lega I, Maraschini A, D'Aloja P, Andreozzi S, Spettoli D, Giangreco M, Vichi M, Loghi M, Donati S; Regional maternal mortality working group. Maternal suicide in Italy. Arch Womens Ment Health. 2020 Apr;23(2):199-206. doi: 10.1007/s00737-019-00977-1. Epub 2019 May 18. PMID: 31104119.
  • Lysell H, Dahlin M, Viktorin A, Ljungberg E, D'Onofrio BM, Dickman P, Runeson B. Maternal suicide - Register based study of all suicides occurring after delivery in Sweden 1974-2009. PLoS One. 2018 Jan 5;13(1):e0190133. doi: 10.1371/journal.pone.0190133. PMID: 29304045; PMCID: PMC5755764.
  • Palladino CL, Singh V, Campbell J, Flynn H, Gold KJ. Homicide and suicide during the perinatal period: findings from the National Violent Death Reporting System. Obstet Gynecol. 2011 Nov;118(5):1056-1063. doi: 10.1097/AOG.0b013e31823294da. PMID: 22015873; PMCID: PMC3428236.

Author Response

The paper provides interesting information on the issue of perinatal mental health care service in Aichi Prefecture, Japan. It is a well-written and useful summary of the current status of the art by the point of view of the care providers.

We appreciate your valuable comments.

Point 1: I have a single minor suggestion for the authors, who are to be congratulated for their work. The high rate of maternal suicide rate (8.7/100000 births) reported in Tokyo's 23 wards from 2005 to 2014 makes the issue of improvement of perinatal mental health care compelling in the Country. In order to make this point clearer to the international readership of the journal, I would suggest adding some data and references on maternal suicide rate in other countries: e.g.,  

  • Khalifeh H, Hunt IM, Appleby L, Howard LM. Suicide in perinatal and non-perinatal women in contact with psychiatric services: 15 year findings from a UK national inquiry. Lancet Psychiatry. 2016 Mar;3(3):233-42. doi: 10.1016/S2215-0366(16)00003-1. Epub 2016 Jan 16. PMID: 26781366.
  • Lega I, Maraschini A, D'Aloja P, Andreozzi S, Spettoli D, Giangreco M, Vichi M, Loghi M, Donati S; Regional maternal mortality working group. Maternal suicide in Italy. Arch Womens Ment Health. 2020 Apr;23(2):199-206. doi: 10.1007/s00737-019-00977-1. Epub 2019 May 18. PMID: 31104119.
  • Lysell H, Dahlin M, Viktorin A, Ljungberg E, D'Onofrio BM, Dickman P, Runeson B. Maternal suicide - Register based study of all suicides occurring after delivery in Sweden 1974-2009. PLoS One. 2018 Jan 5;13(1):e0190133. doi: 10.1371/journal.pone.0190133. PMID: 29304045; PMCID: PMC5755764.
  • Palladino CL, Singh V, Campbell J, Flynn H, Gold KJ. Homicide and suicide during the perinatal period: findings from the National Violent Death Reporting System. Obstet Gynecol. 2011 Nov;118(5):1056-1063. doi: 10.1097/AOG.0b013e31823294da. PMID: 22015873; PMCID: PMC3428236.

Response 1:   We appreciate your valuable suggestion. As you suggested, we cited those papers and added the comments in the Introduction section. But, unfortunately, we could not find the data per 100 000 births in UK and Sweden in those papers, and we showed the data of Italy and the USA.

Reviewer 4 Report

Dear Authors

Thank you for sharing this important work.

I suggest if you add future directions for other researchers 

Author Response

Point 1: Thank you for sharing this important work.

I suggest if you add future directions for other researchers. 

Response 1:Thank you for your helpful suggestion. We added the comment of future directions in the Discussion section.

Round 2

Reviewer 1 Report

the manuscript can be accepted in this form

Reviewer 2 Report

This still remains a very weak paper.